# A Locally Advanced Endometrioid Adenocarcinoma Arising from Vaginal Endometriosis: Management and Review of the Literature

**Mariangela Costanza [1], Fernanda Herrera [1] , Delfyne Hastir [2], Patrice Mathevet [3] and Apostolos Sarivalasis [1],***

[1]  Department of Oncology, Centre Hospitalier Universitaire Vaudois, University of Lausanne,
     1011 Lausanne, Switzerland; Mariangela.Costanza@chuv.ch (M.C.); Fernanda.Herrera@chuv.ch (F.H.)
[2]  Department of Pathology, Centre Hospitalier Universitaire Vaudois, University of Lausanne,
     1011 Lausanne, Switzerland; Delfyne.Hastir@chuv.ch
[3]  Department of Gynecology, Centre Hospitalier Universitaire Vaudois, University of Lausanne,
     1011 Lausanne, Switzerland; Patrice.Mathevet@chuv.ch
*    Correspondence: Apostolos.sarivalasis@chuv.ch; Tel.: +41-79-556-73-62

**Abstract:** Endometrioid adenocarcinoma associated with endometriosis at extrauterine or extraovarian localization is a rare entity. Often presenting with local spread without nodal and distant metastasis, this entity has no specific staging system nor treatment guidelines. In the case of nodal and distant spread, the treatment decision requires personalization. In this article, we present the diagnosis and surgical and systemic treatment of a 56-year-old woman diagnosed with an endometriosis-associated advanced endometrioid adenocarcinoma of the vagina with nodal involvement. Following an extensive review of the scarce data reported to guide the treatment choices in this rare setting, we proposed a multidisciplinary treatment with laparoscopic surgical cytoreduction, four cycles of adjuvant chemotherapy with carboplatin and paclitaxel, and radiotherapy with brachytherapy. Due to an anaphylactic reaction on the first administration, paclitaxel was replaced with nab-paclitaxel. Despite many negative prognostic factors, the patient is free from relapse after 48 months. We report the case of a locally advanced endometrioid adenocarcinoma associated with endometriosis of the vagina, with pelvic nodal spread, and the relevant literature review of similar cases.

**Keywords:** endometriosis-associated cancer; endometrioid adenocarcinoma; nodal metastasis; nab-paclitaxel; surgery; chemotherapy; radiotherapy





## 1. Introduction

Endometrioid adenocarcinoma accounts for 80% of all uterine carcinomas [1]. This histological subtype is also encountered as a type of ovarian adenocarcinoma. Other rare subtypes of endometrioid adenocarcinoma include synchronous uterine and ovarian tumors, and tumors associated with malignant transformation of endometriosis.

Endometriosis is the abnormal growth of benign endometrial tissue outside the uterus, which can affect any pelvic, abdominal, and extra abdominal structure, but it is often observed on the ovaries. The most common sites of extraovarian involvement are the rectovaginal septum, the colon, and the vagina [2]. Endometriosis is a benign condition. Its prevalence is estimated at about 6–10% among women of reproductive age, and up to 50% among women with infertility [3,4]. In approximately 1% of all ovarian endometriosis cases, a malignant transformation can be observed, often resulting in a clear cell or endometrioid adenocarcinoma [5]. The malignant transformation of extra ovarian sites usually occurs in the rectovaginal septum [6,7].

In 1925, Sampson was the first to report a malignant transformation of ovarian endometriosis [8]. He defined the following histopathological criteria, still in use, to diagnose a malignancy arising from endometriosis: (1) coexistence of benign and cancerous tissue in

the same organ; (2) demonstration that the cancer arises from the same tissue; (3) finding of endometrial stroma surrounding characteristic glands. A fourth criteria, the microscopic evidence of a continuum between benign and malign tissue, was added by Scott in 1953 [9].

In this article, we report the case of a woman with endometrioid adenocarcinoma arising from endometriosis of the vaginal fornix with pelvic nodal involvement and an extension to the Douglas pouch and rectal wall. Guidelines for this rare, often misdiagnosed disease are missing.

## 2. Materials and Methods

A 56-year-old woman was admitted to the gynecological emergency department for a 5-month history of progressive perineal discomfort and vaginal serous discharge. The patient had no history of diethylstilbestrol exposure. Her medical history was irrelevant, except for a bilateral oophorectomy for ovarian endometriosis at age 42 followed by 10 years of hormonal substitution.

Upon the gynecological examination, a 2 cm solid lesion of the posterior upper wall of the vagina was detected. The lesion proved to be a grade 1 endometrioid adenocarcinoma on the biopsy specimen. The radiological staging included an intravaginal ultrasound, pelvic magnetic resonance imaging (MRI), and positron emission tomography–computed tomography (PET-CT). A solid lobulated heterogeneous mass of the posterior vaginal fornix measuring 38 mm × 38 mm × 31 mm infiltrating the Douglas pouch and possibly the rectum was detected (Figure 1A–C). No lymph node nor distant metastasis were found.

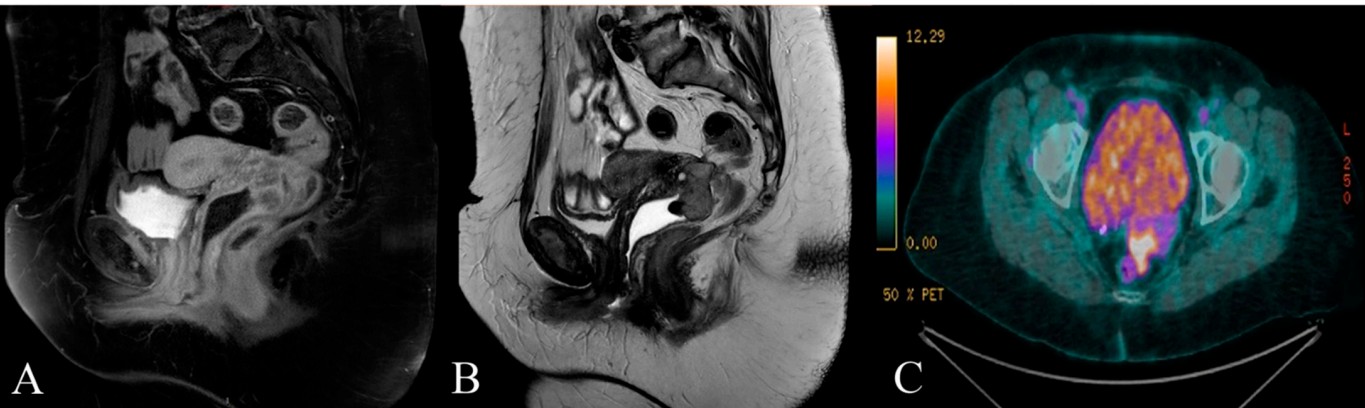

**Figure 1.** (**A**) T1 and (**B**) T2 weighted sagittal magnetic resonance (MR) image of the patient showing the solid mass of the posterior vaginal fornix infiltrating the Douglas pouch and possibly the rectal wall; (**C**) transversal image from positron emission tomography–computed tomography (PET-CT) showing the abnormal fluorine-18-deoxyglucose accumulation in the mass.

The laboratory workout demonstrated normal values on the complete blood count and liver and kidney function. The cancer antigen 125 (CA 125) was elevated at 60 kU/L (upper normal value < 35 kU/L). The carcinoembryonic antigen (CEA) was negative at 1.1 µg/L (upper normal value < 5 µg/L).

The first step in the patient treatment consisted of a laparoscopic surgical debulking. The debulking included a radical total hysterectomy with a bilateral pelvic lymph node dissection, an upper vaginectomy with vaginal reconstruction, an appendicectomy, and a posterior pelvic exenteration with a colon resection and ileostomy. The peritoneal assessment was negative for a tumor. The debulking was negative for macroscopically visible residual disease. No acute surgical complication was observed.

On the pathological examination, the surgical specimen consisted of a solid and cystic tumor of 3 × 2.2 × 3.5 cm. The tumor was infiltrating the vagina's posterior fornix with a 2 cm extension into the posterior and lateral wall of the cervix. The tumor also invaded the rectovaginal septum and the rectal wall. The peritoneal cytology was negative for tumoral cells. On microscopy, a grade 1 endometrioid adenocarcinoma sided with an extensive

endometriosis localization in the resected vaginal mucosa (Figure 2). The adenocarcinoma extended and invaded the rectal wall and the para-rectal and perineal adipose tissue. One out of the fifteen (1/15) resected pelvic nodes was found to be infiltrated by tumor cells. The tumor cells were marked positive for CK7 and focally positive for estrogen receptor (ER) and p16. The staining for CK20 and WT1 was negative. A mismatch repair (MMR) protein immunohistochemistry stain confirmed a proficient tumor.

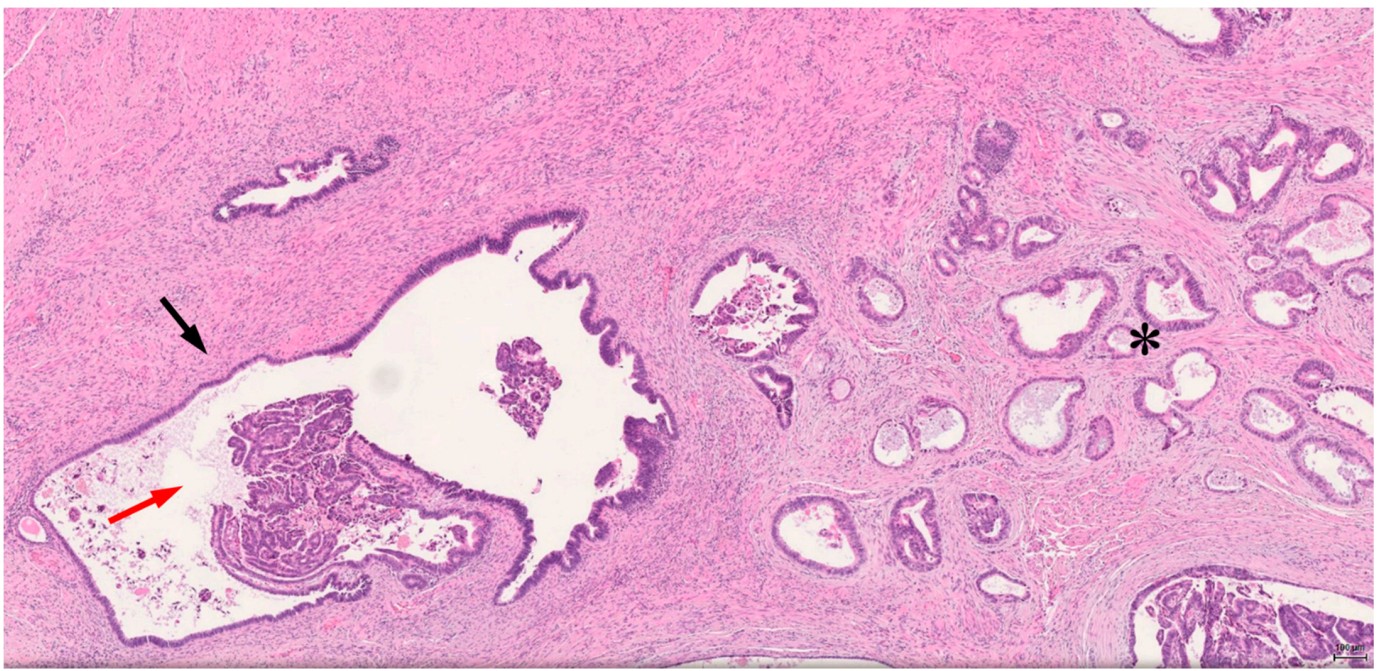

**Figure 2.** Histology of the resected tissue. Transition between cystic endometrioid gland with bland cytologic atypia (black arrow), and malignant transformation of atypical endometriosis (red arrow) and invasive endometrioid adenocarcinoma (*). (HE, ×40).

### 3. Results

We performed a comprehensive literature review on Scopus® and Pubmed® using the key words: "endometrioid carcinoma", "endometriosis", "local invasion", "nodal metastasis", and "adjuvant treatment". The literature review concluded that the vagina is rarely reported as the primary localization of endometrioid adenocarcinoma with only six cases published in the last 20 years (Table 1) [10–13]. Therefore, we extended our research to endometrioid adenocarcinomas arising from endometriosis loci of other abdominal localizations. Of the 13 published cases [14–26], three presented with nodal involvement [14,21,25] and 8/13 had documented adjuvant treatment, including chemotherapy and/or radiotherapy [14,16–20,24,26]. The reported relapse-free survival extended up to 8 years for patients without lymph node involvement. The reported relapse-free survival of the three cases with lymph node metastasis was shorter. Among patients receiving adjuvant treatment, the treatment sequence and regimen was not uniform (Table 2). A sequential adjuvant treatment was proposed because of the local extent of the disease, its aggressive features, and the nodal involvement. The proposed treatment plan consisted of 4–6 cycles of carboplatin AUC5-6 in combination with paclitaxel 175 mg/m$^2$, followed by external beam radiation therapy and vaginal brachytherapy.



**Table 1.** Literature research of vaginal endometrioid adenocarcinoma arising from endometriosis.

| Author | Grade | Location | Size (cm) | Age | Menopause | Nodes | Treatment | Evolution (No Recurrence) |
|---|---|---|---|---|---|---|---|---|
| Lim et al. 2009 [10] | 2 | 1/3 anterior vaginal wall, urethrovaginal septum | 4.3 × 4.2 | 61 | na | N0 | Anterior pelvic exenteration, BSO, pelvic lymph node dissection, urostomy with ileal conduit | 24 months |
| Fruscio et al. 2008 [11] | na | Upper third of the left vaginal wall | 3 × 3.5 | 40 | No | N0 | Neo-adjuvant chemotherapy, radical hysterectomy, BSO, bilateral pelvic and lomboaortic lymph node sampling, resection of left vaginal wall | 24 months |
| Lavery and Gillmer 2001 [12] | na | Vaginal vault | 5 × 3 | 50 | Yes | na | Upper vaginectomy, colon resection (8 cm), and reanastomosis | na |
| Lavery and Gillmer 2001 [12] | na | Left side of vaginal vault | na | 53 | Yes | na | None (non debulkable) | na |
| Fishman et al. 1996 [13] | na | na | na | 47 | Yes | na | Posterior exenteration and RT | 8 years |
| Fishman et al. 1996 [13] | na | na | na | 45 | Yes | na | Radical upper vaginectomy | 7 years |
| Haskel et al. 1988 [27] | na | Upper posterior vagina | na | 53 | Yes | N+ | TAH, vaginectomy, BSO, pelvic and para-aortic lymph node sampling, omental biopsy, partial colpectomy and RT | >24 months |
| Orr et al. 1989 [28] | na | Upper vagina | 6 × 4 × 4 | 60 | Yes | N+ | Resection with low rectal anastomosis, pelvic lymph node dissection, paraaortic node biopsy, and omentectomy | 6 months |
| Granai et al. 1984 [29] | na | Right vaginal apex | na | 47 | Yes | na | Laparotomy (enterolysis), RT, and hormonal therapy | 20 months |
| Kapp et al. 1982 [30] | nc | Posterior vaginal wall | 3 × 4 | 35 | Yes | na | RT | 7 years |
| Hyman 1977 [31] | na | na | na | 50 | na | na | None | na |
| Decelle 1969 [32] | na | na | na | 56 | na | N+ | TAH, BSO, nodal sampling, cystectomy | NA |
| Bamford 1967 [33] | na | Right posterolateral mid-third of vaginal wall | na | 42 | No | na | Hysterovaginectomy | 5 months |

BSO = bilateral salpingo-oophorectomy; RT = radiation therapy; TAH = total abdominal hysterectomy; na = not available.

**Table 2.** Literature research of extraovarian and extrauterine endometrioid adenocarcinoma arising from endometriosis.

| Author | Location | Size (cm) | Age | Menopause | Nodes | Surgery | Adjuvant Chemotherapy | Evolution (No Recurrence) |
|---|---|---|---|---|---|---|---|---|
| Okimura et al. 2018 [14] | Diaphragmatic | 2.4 × 1.6 | 59 | Yes | N+ | Partial resection of diaphragm and liver, BSO, partial omentectomy | 6 cy of carboplatin-paclitaxel | 6 months |
| Lee et al. 2017 [15] | Left paracolic area above the infundibulo-pelvic ligament | 10 × 10 × 8 | 53 | Yes | na | Hysterectomy, BSO, pelvic/para-aortic lymph node dissection, omentectomy | No | 26 months |
| Palla et al. 2017 [22] | Sigmoid colon | 6 | 73 | Yes | na | Sigmoidectomy | No | na |
| Ogi et al. 2016 [23] | Small intestine (ileum) | 6.5 × 4 | 55 | Yes | na | Partial small intestinal resection | No | 5 years |
| Jaiman et al. 2015 [24] | Recto-uterine pouch adherent to the right broad ligament and pelvic wall | 8.8 × 6.5 | 45 | na | na | Hysterectomy, BSO, and mass resection | 6 cy chemotherapy | na |
| Makihara et al. 2015 [25] | Small intestine (ileal mesentery) | 9.5 × 5.5 × 5 | 25 | No | N+ | Partial small intestinal resection | Yes: refusal | 10 months |
| Tarumi et al. 2015 [26] | Bladder | 2.3 | 45 | na | na | Total laparoscopic hysterectomy, partial bladder resection, insertion of bilateral ureteral stents; BO and omentectomy refused by patient | 6 cy docetaxel—carboplatin | 10 months |
| Bawazeer et al. 2014 [16] | Pelvic-abdominal | 16.7 × 10 × 14 | 53 | na | na | Mass excision | 6 cy carboplatin every 3 weeks | End of 6 cy |
| Caballero et al. 2013 [17] | na | 6.5 × 5.6 × 6.8 | 39 | na | na | Rectal anterior resection, hysterectomy, BO | chemotherapy | 1 year |
| Micha et al. 2011 [18] | Pelvic; sigmoid colon, left ureter | 9 cm tumor in the pelvis | 52 | Yes | N0 | Debulking with sigmoid colon resection with low rectal anastomosis, pelvic tumor debulking, bilateral pelvic ureterolysis, lymphadenectomy, appendicectomy, and omental biopsy | cisplatin, RT, and tamoxifen | 5 years |
| Agrawal et al. 2009 [19] | na | na | 50 | na | na | Resection of right anterior chest wall, partial resection of the diaphragm | 6 cy paclitaxel-carboplatin, megestrol acetate | 29 months |
| Park et al. 2009 [20] | Uterine cervix | 3 × 2.2 × 2.2 | 48 | No | na | TAH, BSO | 6 cy cyclophosphamide—cisplatin | 24 months |
| Susumu et al. 2005 [21] | Mesenterium of the sigmoid colon | 6 × 5 × 5 | 62 | Yes | N+ | Sigmoidectomy and lymph node resection | na | 28 months |

BSO = bilateral salpingo-oophorectomy; RT = radiation therapy; TAH = total abdominal hysterectomy; na = not available.

An anaphylactic life-threatening reaction to paclitaxel at the first administration of carboplatin AUC5 and paclitaxel 175 mg/m$^2$ imposed the modification of the initial treatment plan. Because of accumulating evidence in favor of the nab-paclitaxel use in gynecological malignancies, since the second cycle we prescribed a weekly administration of nab-paclitaxel at 100 mg/m$^2$. The treatment was mostly associated with alopecia and hematological toxicity. Severe anemia and neutropenia imposed the treatment interruption after four cycles. We subsequently delivered pelvic radiotherapy (45 Gy in 25 fractions of 1.8 Gy per fraction), which was followed by two brachytherapy sessions of 5 Gy each.

Four years from the completion of her treatment, the patient is free of recurrence.

## 4. Discussion

Endometriosis-associated endometrial adenocarcinoma is a rare gynecological tumor, usually affecting the ovaries [34]. There is debate over its classification, staging, and management. Some experts consider it a form of ovarian cancer, irrespective of its site of presentation, while others consider it a distinct form of uterine cancer. Indeed, endometrioid and clear cell adenocarcinoma of the ovary are often associated with endometriosis. These tumors are frequently low-grade carcinomas, localized to the pelvis, with a favorable prognosis, especially in the early stages and with a 5-year overall survival of 82% [35]. However, locally advanced diseases are at a high risk of relapse. A careful radiological and surgical staging is therefore of paramount importance in order to establish the risk of relapse and the need for adjuvant treatment [35,36]. Epithelial ovarian cancer treatment is well defined [37].

In our patient, the vaginal endometriosis-associated endometrioid adenocarcinoma presented several factors at a high risk of relapse. Thus, standard ovarian carcinoma treatment algorithms could not apply.

In this rare gynecological oncology setting, multidisciplinary tumor board expertise is mandatory to avoid both under- and overtreatment of the patients and provide optimal care [38]. The tumor board of the center of gynecological tumors includes in general a gynecological surgeon, a medical and a clinical oncologist, the specialized pathologist, the radiologist, and a nuclear medicine specialist. After a multidisciplinary discussion, we proceeded to a comprehensive review of the literature. The literature review of similar cases provided evidence for nodal involvement to be a poor prognostic factor. Adjuvant treatment seems to improve the outcome among node-positive endometriosis-associated endometrial adenocarcinomas (Table 2). Due to the lack of a standard adjuvant treatment used in the reviewed cases, including radiotherapy, and chemotherapy as monotherapy or in sequence, we decided to treat our patient as if she was suffering a FIGO IIIC1/IVA high-risk uterine cancer [39].

The patient began her treatment with the standard three weekly carboplatin AUC 5 and paclitaxel 175 mg/m$^2$ regimen. Following a life-threatening anaphylactic reaction on cycle one, day one, her treatment plan was modified for continuous weekly nab-paclitaxel at 100 mg/m$^2$ (Supplementary Material, Figure S1). Those allergic reactions are a well-known side effect of standard chromophore-diluted paclitaxel, which occur in about 10% of patients [40]. The nab-paclitaxel was preferred to docetaxel because of the availability and accumulating evidence on safety and efficacy for the use of nab-paclitaxel in gynecological cancers [40,41]. Her treatment was well tolerated with mostly hematological toxicities requiring G-CSF stimulation (Supplementary Material, Table S1). After completing four cycles of chemotherapy, the patient received radiotherapy due to the high-risk setting, in accordance with the PORTEC-1 [42] and GOG99 [43] trials. The patient was able to receive three weekly cycles, and despite hematological toxicity, her treatment was carried out without interruption. At four years of follow up, the patient is in complete remission (CR).

## 5. Conclusions

In summary, we report the case of a locally advanced endometrioid adenocarcinoma arising from vaginal endometriosis. The need for an adjuvant treatment for ectopic lesions

has to be personalized according to the disease extension and the presence of risk factors of relapse. Although we recommend a standard adjuvant regimen as per uterine cancer, the treatment should be adapted to the patient's tolerance.

**Supplementary Materials:** The following are available online at https://www.mdpi.com/article/10.3390/reports4030029/s1, Figure S1. Adjuvant treatment, Table S1. Treatment-related adverse events.

**Author Contributions:** Conceptualization and drafting, M.C. and A.S.; methodology, A.S.; software, M.C.; validation M.C., F.H., D.H., P.M. and A.S.; investigation, A.S.; writing—original draft preparation, M.C. and A.S.; writing—review and editing, M.C. and A.S.; supervision, A.S.; project administration, A.S. All authors have read and agreed to the published version of the manuscript.

**Funding:** This research received no external funding.

**Institutional Review Board Statement:** Not applicable.

**Informed Consent Statement:** Informed consent was obtained from all subjects involved in the study. Written informed consent has been obtained from the patient(s) to publish this paper.

**Data Availability Statement:** Not applicable.

**Conflicts of Interest:** The authors declare no conflict of interest.

## Abbreviations

| | |
|---|---|
| AUC | area under the curve |
| BSO | bilateral salpingo-oophorectomy |
| BO | bilateral oophorectomy |
| cy | cycles |
| CA 125 | cancer antigen 125 |
| CEA | carcinoembryonic antigen |
| MRI | magnetic resonance imaging |
| PET-CT | positron emission tomography–computed tomography |
| RT | radiation therapy |
| TAH | total abdominal hysterectomy |
| Gy | gray |
| na | not available |
| G-CSF | granulocyte-colony stimulating factor |
| CR | Complete response |

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
