# Peer review of "A Locally Advanced Endometrioid Adenocarcinoma Arising from Vaginal Endometriosis: Management and Review of the Literature"

_reports, doi:10.3390/reports4030029_

Round 1

Reviewer 1 Report

This case report is potentially interesting and the authors investigated the reports about extraovarian and extrauterine endometrioid adenocarcinoma. However, I think that the content is confusing due to inappropriate subheadings, such as “Materials and Methods,” and “Results.” Therefore, I think it is better to change to the general case report format and describe the previous reports in the discussion. Moreover, in Figure 1, I think it is better to point out the lesion with an arrow, which will help readers for a better understanding.

Author Response

About: Manuscript ID: reports-1370685 - Revision

Dear Editor,

We thank the reviewers for their invaluable help and useful comments to improve our manuscript.

Please find here-bellow our answer to the reviewer

Reviewer 1

  1. We thank the reviewer for his remarks about the confusion he/she is highlighting as an effect of the inappropriate subheadings used in the manuscript and his suggestion to change the subheading as per the general case report format. The subheading and structure used in our manuscript was the one provided by the journal template and our manuscript was edited accordingly. If the editor agrees, we could adapt the subheading to make them more intuitive but this is not just a case report but also a review of the literature of this rare oncological situation.

  1. We thank the reviewer for his suggestion about figure 1. We have updated the figure one and added 3 arrows to help the readers better understand the tumor localization. This updated version is called figure 1bis.

Reviewer 2 Report

General

The authors present a well written and interesting case report and literature survey about a case of endometroid adenocarcinoma arising at endometriosis of the vaginal wall. There are only minor questions arising.

Specific

Page 3 : “one part of the fifteen …..”

               Show exact number.

Table 2: Use uniformly “appendectomy” (Citation 18) or                        

               “appendicectomy” (page 2 ,line 31).  

Page 3: Was there a difference of the immunohistochemical staining

               Pattern between the preexisting endometriosis and the

               Endometroid adenocarcinoma?

Page 2: 4th paragraph: Biopsy revealed a grade 2 adenocarcinoma. On

              3, 3rd line the lesion is diagnosed as grade 1 carcinoma.

Author Response

About: Manuscript ID: reports-1370685 - Revision

Dear Editor,

We thank the reviewers for their invaluable help and useful comments to improve our manuscript.

Please find here-bellow our answer to the reviewer

Reviewer 2

I. We thank the reviewer for his suggestion on page 3; an adaptation was made on the updated manuscript.

II. We thank the reviewer for his suggestion about the “appendicectomy “, an adaptation was made on the updated manuscript.

III. We thank the reviewer for his query about differences in the immunohistochemical staining pattern between endometriosis and EC. The initial pathology report did not reported on the IHC staining of endometriosis in comparison to the one of the EC. Thus, if it is required to add the differences in the IHC profile between the Endometriosis and EC an additional pathological examination is require. The reason we did not performed earlier was that we considered this information beyond the scope of this paper. In the updated manuscript, we have included the Figure 2, a hematoxylin-eosin stain of the EC within the endometriosis to present the localization and microscopic differences between the two pathologies. We have also updated the manuscript paragraph about the IHC staining of the EC.

IV. We thank the reviewer for his correction, indeed it was a grade1 tumor.

Reviewer 3 Report

The authors present a case report with ancillary literature review about a locally advanced endometrioid adenocarcinoma arising from vaginal endometriosis.

The topic is by far interesting because there are no share guidelines to guide the best management.

The literature review is methodologically well perfomed and it is an addition to the paper.

I have some points which I would like the authors to address more in detail:

1) Was the case discussed in a multidisicplanry setting? If so which were the specialists involved? I recommend that the role of multisdisciplinary approach in such rare cases should be underlined more clearly in the text also addinf some relavant literature about the value of multidisplinary approach in rare cancers.

2) How was the therapeutic strategy followed? Was additional chemotherapy or radiotherapy administered? how long is the follow-yp and with which associated disease free survival? The authors should provide more details about the case.

Author Response

About: Manuscript ID: reports-1370685 - Revision

Dear Editor,

We thank the reviewers for their invaluable help and useful comments to improve our manuscript.

Please find here-bellow our answer to the reviewer

Reviewer 3

I. We thank the reviewer for his remarks on the multidisciplinairy setting. Indeed all gynecological oncology patient-files are discussed in our Center’s Tumorboard. Thus, we added a paragraph to the discussion of our updated manuscript and some relevant literature highlighting the benefit of this approach in this setting.

II.  We thank the reviewer for his remarks on the therapeutic scheme and follow up. The patient was treated according to table 1 of the supplementary material with adjuvant chemotherapy, radiotherapy and brachytherapy. She is in CR after 4 years of FU. The therapeutic strategy and adjuvant treatment (chemotherapy, radiotherapy and brachytherapy) is resumed in detail in the last paragraph of the “Results” section.

Round 2

Reviewer 3 Report

The authors hace addressed my previous concerns.

I have no furhter concerns.